# Impact of the COVID-19 Pandemic on Nutrition, Sleep, Physical Activity, and Mood Disorders of Polish Children

**DOI:** 10.3390/nu15081928

**Published:** 2023-04-17

**Authors:** Karolina Krupa-Kotara, Gabriela Wojtas, Mateusz Grajek, Martina Grot, Mateusz Rozmiarek, Agata Wypych-Ślusarska, Klaudia Oleksiuk, Joanna Głogowska-Ligus, Jerzy Słowiński

**Affiliations:** 1Department of Epidemiology, Faculty of Public Health in Bytom, Medical University of Silesia in Katowice, 41-902 Bytom, Poland; awypych@sum.edu.pl (A.W.-Ś.); koleksiuk@sum.edu.pl (K.O.); jglogowska@sum.edu.pl (J.G.-L.); jslowinski@sum.edu.pl (J.S.); 2Student Scientific Society, Department of Epidemiology, Faculty of Public Health in Bytom, Medical University of Silesia in Katowice, 41-902 Bytom, Poland or gabrielawojtas54@gmail.com (G.W.); d201137@365.sum.edu.pl (M.G.); 3Department of Public Health, Department of Public Health Policy, Faculty of Public Health in Bytom, Medical University of Silesia in Katowice, 41-902 Bytom, Poland; mgrajek@sum.edu.pl; 4Doctoral School, Medical University of Silesia in Katowice, 40-055 Katowice, Poland; 5Department of Sports Tourism, Faculty of Physical Culture Sciences, Poznan University of Physical Education, 61-871 Poznan, Poland; rozmiarek@awf.poznan.pl

**Keywords:** COVID-19, SARS-CoV-2, pandemic, children, eating behavior, eating habits, lifestyle

## Abstract

The harmful consequences of the COVID-19 pandemic on children are its impact on eating habits, physical activity, sleep, and mood disorders. In the future, this may result in a higher prevalence of obesity and diet-related diseases. Therefore, this study aimed to assess the impact of the COVID-19 pandemic on children’s eating behavior and lifestyle. The study was conducted using a proprietary questionnaire on dietary and lifestyle habits before and during the pandemic, and the reasons for changes due to the pandemic. The study involved 294 parents of children in grades 1–8 in elementary schools in two regions of Poland. The survey showed that during the pandemic, the percentage of children eating five regular meals daily, including fruits and vegetables, and engaging in daily physical activity decreased. However, the percentage of children spending more than 4 h a day in front of a screen increased (*p* < 0.05). The main reasons for changes in eating habits and physical activity were less eating out, lack of motivation, obstruction, and lack of access to sports facilities (*p* < 0.05). The pandemic had a significant impact on reduced levels of physical activity and increased time spent in front of a screen. In summary, among the reasons for changes in children’s dietary and lifestyle habits, factors related to the pandemic itself, i.e., social restrictions, restrictions, closure of schools and other facilities, and fear of coronavirus infection, had the greatest impact.

## 1. Introduction

In 2019–2020, the world experienced the emergence of a new coronavirus disease—COVID-19 [1,2]. Since the first case of SARS-CoV-2 virus infection was detected in Wuhan, China, the COVID-19 pandemic has rapidly spread to other countries and continents [2,3]. It has had a significant negative impact on the global community, the global economy, and everyday life [3,4].

During a pandemic, children and adolescents mainly act as a host to the SARS-CoV-2 virus, transmitting it into the environment [2]. For the youngest children, the biggest risk associated with COVID-19 is not the issues related to the disease itself, but the accompanying aspects of the disease. These include inadequate nutrition that can lead to both overweight and underweight, reduced levels of education, impact on mental health, social isolation, and lower rates of health care [1]. The prevention and health policy initiative for diet-related diseases is the WHO European Childhood Obesity Surveillance Initiative (COSI). The initiative’s unique system has been estimating overweight and obesity data among school-aged children for more than 10 years. The program consists of taking standard anthropometric measurements (weight and height) among approximately 300,000 children in the WHO European Region. The results will provide representative national and regional data to analyze the determinants of overweight and obesity in children at different latitudes, including Poland. The target group for COSI is the national population of elementary school children aged 6–9. Each country is responsible for collecting and analyzing national data through a pre-designated institute responsible for overall coordination and management at the national level. The data are analyzed both at the national level and by a WHO/Europe team that conducts joint analyses of the datasets between countries. The data management process will be completed with the release of information through reports and/or scientific publications [5].

The pandemic has greatly affected the quality and quantity of foods and products consumed, but also the diversity of the diet [1]. Another important aspect of the pandemic is the loss of jobs, which consequently led to financial constraints. The result has been a shift in the food products purchased to cheaper so-called “low-end” foods. However, not every household experienced the negative change in nutrition associated with the COVID-19 pandemic. Some families saw the situation as an opportunity to cook home-cooked meals and were more likely to buy fresher products [6].

Low physical activity and poor diet potentially increase the risk of excess body weight and diet-related diseases. Therefore, an important part of the public health response during the COVID-19 pandemic is to educate children about healthy eating habits and physical activity [7]. Outdoor activities with limited contact with other people were particularly recommended, as they offered the greatest guarantee of safety [8].

As children experience a change in daily habits, a difference in sleep patterns should also be expected. There is evidence that daily habits, i.e., time spent in front of a screen or a set schedule, greatly affect sleep length and quality. A longer sleep duration is associated with a lower body mass index, higher quality of daily diet, and increased levels of physical activity [2].

The COVID-19 pandemic forced the introduction of remote learning and further increased the time children spent in front of screens. The positive aspects were further educational opportunities and continued contact with peers during social isolation. However, it also had a negative impact, extending sedentary time and increasing the risk of anxiety or depression. Studies conducted during the pandemic showed that children’s time spent in front of screens increased by up to 5 h a day compared to before the pandemic [6].

Studies have shown the negative impact of a long process of social isolation on people’s psychological state, leading to depression, stress, or anxiety in many cases [4]. Blockage and social isolation have affected the population of children and adolescents to a greater extent, more vulnerable to stressful situations, and may consequently negatively affect their adult lives. Stress and anxiety have led to changes in dietary choices. A large proportion found solace and comfort in consuming high-fat or high-energy foods, which consequently led to disorders of normal body weight [9].

However, not enough research has been done on the case of children and adolescents. It is most likely that the increase in the spread of coronavirus has caused a high burden of mood among children and adolescents, which negatively affects their physical and mental development. Therefore, research on the aspect of the younger generation is extremely important, since changes in childhood will cause consequences in adulthood. For children, the biggest risks associated with COVID-19 are not issues related to the disease itself, but the accompanying aspects of the disease. These include inadequate nutrition that can lead to both overweight and underweight, reduced levels of education, impact on mental health, social isolation, and lower rates of health care [10,11].

During the COVID-19 pandemic, the concept of self-care evolved. According to the National Alliance of Mental Health, it is a decision made to improve health that should be practiced every day. The concept includes health, body movement, nutrition, sleep, rest, and physical touch. The role of the child’s family or caregivers in educating the child about health care is crucial, with a focus on eating behaviors. This includes a healthy diet and adherence to proper habits such as personal hygiene [12,13].

The COVID-19 pandemic is assumed to have had negative effects on all areas of Polish children’s lives, particularly those related to lifestyle, to which the lockdown has contributed. Therefore, this study aims to assess the impact of the COVID-19 pandemic on the eating behavior and lifestyle of children attending elementary schools. The objective was pursued by: (1) Assessing changes in children’s body weight due to the COVID-19 pandemic. (2) Assessing the change in children’s eating habits, physical activity, and lifestyle as a result of the COVID-19 pandemic. (3) Assessing the causes of changes in children’s eating behavior and lifestyle due to the COVID-19 pandemic.

## 2. Materials and Methods

### 2.1. Study Area

The survey was conducted among parents of children attending grades 1–8 in elementary schools in Silesia and Lesser Poland, two of the sixteen voivodeships (Poland), the most representative of the country’s structure. Using independent double-masked random sampling, 45 schools were selected out of 2941 elementary schools. Each school had an average of 16 ± 6 classes, from which 294 children were included in the study. A mailing list was set up at the selected elementary schools, and school principals were asked to distribute the author’s questionnaire to parents of students. The necessary sample size was calculated according to the size of the population of elementary school students of the Silesian and Lesser Poland regions. It was estimated that a sample of 294 elementary school students would be sufficient and representative of the Silesian region in Poland. According to the GUS (Central Statistical Office) Report [14], it was assumed that the population of elementary school students in the Silesia and Lesser Poland regions is 740640. The sample size was calculated according to the formula: Nmin = NP ⋅ (α^2^ ⋅ f(1 − f)) ÷ NP ⋅ e^2^ + α^2^ ⋅ f(1 − f), where: Nmin—minimum sample size; NP—the size of the population from which the sample is drawn; α—confidence level for the results; f—the size of the fraction; e—assumed maximum error. A minimum sample size of 138 was calculated for the population of Silesian, and Lesser Poland students (α = 0.95; f = 0.9; e = 0.05). Based on these calculations, the collected group was considered representative.

### 2.2. Inclusion Criteria

The inclusion criteria for the study were: The age of the child in the range of 6–14 yearsA child attending grades 1–8 of elementary schoolWritten consent of the parent expressed through participation in the study.Participation in the study was anonymous and entirely voluntary.

The study complies with the provisions of the Declaration of Helsinki, as amended. The study design, in light of the Act of 5 December 1996, on the professions of physician and dentist (Journal of Laws of 2011, No. 277. item 1634, as amended), is not a medical experiment, as the study focuses on the subjective experiences of the participants. The study was approved by the Bioethics Committee of the Silesian Medical University in Katowice (ID. PCN/CBN/0052/KB/127/22). 

### 2.3. Research Tool

The method used in the study was an online proprietary questionnaire, which was developed based on a validated questionnaire of questionnaire to evaluate the impact of COVID-19 on lifestyle-related behaviors: eating habits, activity, and sleep behavior. Cronbach’s α value of 0.7 or higher indicates good internal consistency [15]. The questionnaire was translated from English to Polish and then reformulated to adapt to the survey in a Polish setting. All questionnaire data were self-reported by parents.

To test the theoretical relevance of the questionnaire, the questions were evaluated by competent judges consisting of a nutritionist, a psychodietitian, and a psychologist. Each judge was given a questionnaire to assess the degree of comprehensibility of each question. In addition, the psychologist assessed the extent to which each question examined changes in particular life spheres, as well as whether it examined any other variables. In addition, the questions were evaluated for comprehensibility by ten randomly selected adults. Most of the questions were rated positively by the judges. Questions that were judged by at least one judge to be incomprehensible, or to examine other variables, were changed or removed. To assess the reproducibility of the results obtained with the questionnaire used, the value of the χ parameter (Cohen’s kappa) was calculated for each questionnaire question (results obtained in the baseline and retest). A total of 74.8% of the questions had an excellent agreement <0.75.

The survey questionnaire consisted of a metric, a section on children’s lifestyles before the COVID-19 pandemic, a section on children’s lifestyles during the COVID-19 pandemic, and a section covering the reasons for the changes that occurred as a result of the pandemic. The metric questions included socio-demographic information: year of birth, gender, place of residence, and the class the child attends, as well as anthropometric data, i.e., the child’s weight [kg] and height [cm]. The data came from self-reporting by the child’s parents. Based on weight and height, BMI was calculated, which was then cross-referenced to the OLA and OLAF centile grids. The following cut-off points were adopted: underweight—less than the 5th percentile; healthy weight—5th percentile to less than the 85th percentile; overweight—85th percentile to less than the 95th percentile; obesity—95th percentile or greater. Values of the 3rd and 97th percentile are taken as the lower and upper limits of normal. This is because this limit is closest to the statistical norm, that is, the mean value ± 2SD. It should be emphasized that the 3–10 and 90–97 centile intervals (channel) should be considered as the borderline of the norm, requiring observation or follow-up clinical examinations of the child whose height and weight values are located there (especially the 3–10th centile) [16] to analyze anthropometric parameters. The questions in the following sections of the questionnaire concerned eating habits, daily activities, i.e., physical activity, sleep, and time spent in front of a screen, as well as children’s well-being.

### 2.4. Statistical Analysis

To analyze the results, a database was created in MS Excel 2016. To examine statistical relationships, a Chi^2^ test was performed. Analyses were carried out using the Kruskal–Wallis test (non-parametric) for between-group analyses of variance, using data verification software. All statistics determinations were made according to Life Cycle Committee guidelines [17]. Analysis was performed using Statistica 13.0 Stat Soft Polska. The results obtained from the questionnaires were presented in the form of tables and figures. Statistical significance was determined at *p* < 0.05.

## 3. Results

### 3.1. Socio-Demographic Information

Table 1 shows the baseline characteristics of the study group. The study included 294 parents of children aged 6–14 attending elementary school, including 149 girls and 145 boys. The average age of the children was 11.49 ± 2.35 years. The largest group of respondents were children from the youngest grades of elementary school (36.7%). More than half of the children surveyed (52.7%) lived in the city. (Table 1). The structure of children and adolescents in Poland is the same as in our sample [14].

### 3.2. BMI and Weight Changes in Children during the COVID-19 Pandemic

Based on an analysis of BMI about age based on centile grids, the vast majority of girls (69.1%) and boys (66.2%) were of normal weight. More than 20% of girls were underweight (21.5%), and almost 10% were overweight (9.4%). Boys were more likely to be overweight (17.2%) compared to underweight (15.2%) (Figure 1).

For both girls and boys, the majority of parents reported no change in weight—50.3% of girls and 54.5% of boys. Approx. one-third of the parents stated that their children’s weight had increased—33.6% of girls and 32.4% of boys (Figure 2).

### 3.3. Comparisons of Children’s Dietary and Lifestyle Behaviors before and during the COVID-19 Pandemic

The highest percentage of the level of consumption of five regular meals per day was the frequency—daily before the pandemic and during the pandemic (N = 126–79.75%). Statistical analysis showed that there was a relationship between the frequency of consumption of five regular meals per day before the pandemic and during the pandemic, T = 13.592, r = 0.77, *p* < 0.001 (Table 2).

In terms of children’s eating habits, compared to the pre-pandemic period, the percentage of children’s daily consumption of vegetables and fruits decreased during the pandemic (39.5% and 33.3%, respectively). During the pandemic, the percentage of children consuming fruits and vegetables less frequently, i.e., 3–4 times a week (26.2% and 30.6%, respectively) and 1–2 times a week (13.3% and 16.0%, respectively) also increased Statistical analysis showed that there was a correlation between the frequency of fruit and vegetable consumption before the pandemic and during the pandemic, T = 10.947, r = 0.872, *p* < 0.001 (Table 3).

Compared to the pre-pandemic period, during the pandemic, the percentage of consumption of salty snacks 1–2 times a week decreased (65.6% and 55.4%, respectively), while the percentage of more frequent consumption increased—3–4 times a week (16.3% and 22.8%, respectively) and 5–6 times a week (2.0% and 4.8%, respectively). Statistical analysis showed that there was a correlation between the frequency of consumption of unhealthy snacks before and during the pandemic, T = 11.234, r = 0.673, *p* < 0.001 (Table 4).

In terms of physical activity, compared to the pre-pandemic period, the percentage of frequency of doing a 30 min activity 1–2 times a week increased during the pandemic (27.9% and 40.5%, respectively). However, the percentage of frequency of 5–6 times a week (12.9% and 9.5%, respectively) and daily (17.0% and 10.5%, respectively) decreased Statistical analysis showed that there was a correlation between engaging in moderate-intensity exercise/sports before and during the pandemic, T = 10.625, r = 0.722, *p* < 0.001 (Table 5).

In terms of time spent in front of a screen, compared to the pre-pandemic period, the percentage of children spending 1–2 h in front of a screen decreased significantly during the pandemic (46.6% and 26.9%, respectively), while the percentage of children spending 2–4 h (27.6% and 37.1%, respectively) and more than 4 h (5.1% and 25.5%, respectively) increased. Statistical analysis showed that there was a correlation between the length of time spent in front of a TV/phone/computer screen before and during the pandemic, T = 12.331, r = 0.577, *p* < 0.001 (Table 6).

In the context of children’s sleep quality ratings, compared to the pre-pandemic period, the percentage of sleep quality ratings of “very good” (61.9% and 39.8%, respectively) decreased during the pandemic, while the percentage of sleep quality ratings of “good” (37.1% and 52.0%, respectively) and “bad” (0.3% and 7.5%, respectively) increased. Statistical analysis showed that there was a correlation between the length of time spent sleeping before and during the pandemic, T = 10.232, r = 0.671, *p* < 0.001 (Table 7).

In terms of children’s feelings of stress or anxiety, the percentage of children feeling no stress/anxiety at all (27.2% and 17.3%, respectively) or sometimes feeling stress/anxiety (68.0% and 63.6%, respectively) decreased during the pandemic compared to the pre-pandemic period. However, the percentage of children feeling stress/anxiety often (3.7% and 15.6%, respectively) or very often (1.0% and 3.4%, respectively) increased. Statistical analysis showed that there was a correlation between the degree of feeling stress/anxiety before and during the pandemic, T = 12.011, r = 0.781, *p* < 0.001 (Table 8).

### 3.4. Causes of Changes in Children’s Eating Behavior and Lifestyle as a Result of the COVID-19 Pandemic

The largest percentage of parents (57.1%) stated that their child’s nutrition had not changed much. The most common reasons for the change were less eating out (19.7%), more available time for cooking (16.3%), and higher cost of groceries (16.0%) (Figure 3).

More than half of parents (53.7%) stated that their children’s physical activity had not changed. As reasons for the change, the largest percentage of parents indicated a lack of motivation (21.8%), restrictions in parks and public places (19.7%), and lack of access to sports facilities and gyms (19.7%) (Figure 4).

## 4. Discussion

The study conducted concerns the impact of the COVID-19 pandemic on children’s eating habits and elements of their daily lifestyles, i.e., sleep, physical activity, or time spent in front of a computer or phone screen. It is worth emphasizing here the importance of the consequences of the pandemic condition, which may impact children for the next years of their lives. The COVID-19 pandemic has had many negative effects on vulnerable groups, including children. Its effects may be felt for many years to come. Therefore, research on trends in the impact of the pandemic on various areas of young people’s lives seems particularly important. Many times, they overlap with already negative health trends such as the burden of chronic non-communicable diseases. Such research will allow a better definition of the health needs of the study populations and proper targeting of prevention, case management, and social support groups, and thus more effective assistance and reduction of social and economic costs and prevention of social inequalities in health.

A key issue is a role the pandemic has played in the weight aspect of children. In a study conducted, parents noticed an increase in weight in one-third of the children. This may have been caused by a deterioration in eating habits, but also to a large extent by a decrease in the level of physical activity in some children and an increase in the amount of time spent sitting. Similar results were reached in their study by Azrak et al. [18], in which parents declared an increase in their children’s weight during the pandemic; moreover, weight gain continued to increase in boys even when lockdown measurements were relaxed, although sedentary behavior decreased and quality of life improved, indicating that the effects of pandemic lockdown may be difficult to reverse. A comparable result was also obtained by Androutsos et al. [19] in a study conducted in a group of Greek children aged 2–18, where 35% of them had their parents report an increase in weight during the pandemic. On the other hand, an Italian study by Pujia et al. [20], showed that as many as 59.7% of children aged 5–14 had weight gain during the pandemic.

During the pandemic, there were many factors affecting children’s diets. The daily routine was disrupted, some children were deprived of the opportunity to enjoy school meals, and some used large amounts of free time to shop independently or eat out. The survey showed a decrease in the percentage of children eating vegetables and fruits daily (39.5% vs. 33.0%). These are essential foods that should mandatorily prevail in the daily diet. Sharma et al. [21] surveyed more than a thousand American families with children characterized by low income, and 41.4% of households showed a decrease in the supply of fruits and vegetables. On the other hand, interesting results were obtained by Kolota and Glabska [22], who found that, during the pandemic, in a group of 1334 children aged 10–16, there was an increase in the percentage of children consuming more than four servings of vegetables per day (7.5% vs. 11.1%) and more than three servings of fruit per day (19.0% vs. 27.4%). Moreover, in an Italian study [13], the authors noted an increase in fruit and vegetable intake in 19.0% of children aged 5–14 years surveyed. More time spent at home has allowed many families to prepare home-cooked meals together and thus maintain proper eating habits.

The study showed an increase in the percentage of children consuming salty snacks 5–6 times a week (2.0% vs. 4.8%) and daily (3.1% vs. 4.8%). Koletzko et al. [23] surveyed 1000 German parents of children under the age of 14 and showed that during the pandemic, 18% of them noticed an increase in their children’s consumption of salty snacks. A Greek study by Siachpazidou et al. [24] of children aged 4–12 proved that 46.2% of parents admitted that their children maintained unhealthy eating habits during the pandemic.

In a study conducted, almost 20% of parents cited eating out less as a reason for changing the eating behavior of Polish children during the pandemic. Radwan et al. [25] surveyed a Palestinian group of 2398 children and adolescents aged 6–18 and showed that as many as 93.3% of them avoided ordering meals from outside, preferring instead to prepare meals at home. Thus, the pandemic may have had a positive effect on some children and adolescents, who, through more free time, were motivated to prepare meals with their own hands or together with their families.

The pandemic has forced the governments of countries to impose many restrictions and close various types of facilities that are important elements of the daily lives of children and young people, i.e., schools, sports facilities, or even playgrounds. Physical education lessons and opportunities for additional activities have been lacking. The study showed that the percentage of children engaged in half an hour of physical activity every day dropped from 17.0% to 10.5% during the pandemic. In contrast, the percentage of children not doing any activity at all increased (8.8% and 12.9%, respectively). Governments must consider the negative effects that restrictive measures during the COVID-19 pandemic have on children’s physical activity and act to ensure high levels of physical activity. Hourani et al. [26], on the other hand, observed in their study an almost twofold increase in the percentage of children who did not engage in any physical activity during the pandemic—in Jordanian children aged 6–12, this was a change from 16.7% before the pandemic to 29.8% during the pandemic. Furthermore, in a Greek study [18], the authors showed that 53.8% of parents of children aged 4–12 confirmed their low or moderate physical activity during the pandemic. Parents cited the closure of schools and the start of distance learning as reasons for their children’s activity deterioration.

The long time spent at home during pandemic restrictions promoted the excessive use of electronic equipment, i.e., TV, computer, and telephone. The study showed that the percentage of children spending more than 4 h a day in front of a screen increased significantly during the pandemic (5.1% vs. 25.5%). Moreover, in another Polish study, Kolota and Glabska [22] showed an increase in the percentage of children aged 10–16 watching TV for more than 2 h a day (78.3% vs. 88.4%). In another Polish study, Luszczki et al. [27] also proved an increase in the percentage of children aged 6–15 who watched movies or online programs more than 6 h a day on weekends during the pandemic (1.3% vs. 5.1%). In contrast, Fasano et al. [28] showed that more than one-third of the Argentine children surveyed, aged 4–11, spent more than 2 h a day watching movies or programs on their phones or TV. In a Greek study [24], the authors showed that as many as 61.6% of parents indicated the impact of school closures on the increase in the use of electronic devices to watch movies or play computer games. Of course, the increased time spent by children in front of a screen was undoubtedly influenced by remote learning, but most studies included time consumed for entertainment purposes.

In terms of the COVID-19 pandemic, the study proved the importance of stress and anxiety associated with social isolation and long hours spent by children on remote learning, in terms of changing sleep patterns. The study showed that the percentage of children spending less than 6 h of sleep increased during the pandemic period (2.0% and 5.0%, respectively), while the percentage of children spending more than 8 h of sleep decreased (50.0% and 45.0%, respectively). In opposition is a Greek study [24], which showed a decrease in the percentage of children devoting less than 8 h to sleep during the pandemic (15.4% and 4.8%, respectively), while an increase in the percentage of children devoting more than 10 h to sleep was observed (13.3% and 24.2%, respectively). An Argentine study [28] showed that more than half of the children surveyed, aged 4–11 years (54.2%), went to bed at later hours during the pandemic compared to before the pandemic. Ventura et al. [29], in their study of children under the age of 17, showed that nearly 80% of children during the pandemic were characterized by later bedtimes, and nearly 20% had insufficient hours of sleep. Shorter sleep during the pandemic may have been influenced by the long time spent on the phone or playing computer games, while long hours of sleep were favored by school activities from home and not having to prepare for school.

The pandemic also left its mark on children’s mental health to some extent. The survey showed a decrease in the percentage of children who did not feel stress or anxiety at all during the pandemic (27.2% vs. 17.3%) and an increase in the percentage of children who often felt stress or anxiety (3.7% vs. 15.6%). The level of feeling negative emotions may have been influenced by staying at home too long, lack of social interaction, and fear of coronavirus infection. In the survey, 30.6% of parents indicated boredom or loneliness as a reason for the change in stress and anxiety levels. In an Italian study [13], the authors showed that 37% of parents emphasized feelings of loneliness in their children during the pandemic. Similarly, Francisco et al. [30], in their study analyzing a group of Italian, Spanish, and Portuguese children, showed that 33.2% of parents reported feelings of loneliness in their children during the pandemic. In addition, they proved that 27.2% of the children surveyed experienced fear of coronavirus infection. In the study conducted, 23.1% of Polish children feared infection, which shows that the virus itself disturbed the psychological state of children to some extent.

Mental health in children is concerned with their psychological, emotional, and social well-being, particularly the development of healthy relationships within the family and among peers, the improvement of self-esteem, and the ability to cope with stress. Mental health problems, which are increasingly common in young adulthood, carry the risk of continuing into adulthood [29]. During infancy, early childhood, and adolescence, the central nervous system is particularly sensitive to any stressful situations that arise in daily life. This can result in short- and long-term negative consequences on physiological, cognitive, and behavioral grounds. A study conducted in 2020 in India showed that in children and adolescents in quarantine, the prevalence of feelings of anxiety and fear increased. This was mainly due to concerns about the family’s financial issues, or exposure to SARS-CoV-2 and infecting others [30].

Blockage and social isolation have affected the child and adolescent population to a greater extent, more vulnerable to stressful situations, and may consequently negatively affect their adult lives. Stress and anxiety have led to changes in dietary choices. A large proportion found solace and comfort in consuming high-fat or high-energy foods, which consequently led to disorders in normal body weight [31,32]. The association of a stressful environment and the resulting increase in foraging early in life may be contributing to the spread of the obesity epidemic among the younger generation [33].

In studies conducted before the pandemic, it was observed that in the case of parents with diagnosed certain psychiatric disorders, children are more likely to develop psychopathology. Another aspect of the COVID-19 pandemic was that most parents had to adjust their daily lives and work at home to take care of their children while remaining on remote learning. The consequence of such a sudden overload of parents under stressful conditions was an increased risk of children developing emotional and behavioral problems [34,35,36].

Rucińska et al. [10] showed that the lockdown caused by the COVID-19 pandemic generated mainly undesirable lifestyle changes (decreased physical activity, increased screen time); however, desirable effects were also observed (increased meals eaten at home and the amount of sleep).

During the COVID-19 pandemic, children and adolescents are subject to the same nutritional guidelines as adults. The periods of their growth and development are crucial when potential malnutrition can negatively affect their future health. Therefore, it is crucial to provide the growing body with all the necessary nutrients in adequate amounts, paying extra attention to vitamin D. An important aspect to pay attention to in the context of child nutrition is the increase in the prevalence of obesity, which is further exacerbated during the pandemic. For the sake of physical, mental, and social health, the promotion of a properly balanced diet and daily physical activity is crucial [15].

The Committee on Human Nutrition Sciences [15] has developed dietary recommendations for children during the COVID-19 pandemic. A child should drink several glasses of water every day, or about 1–2.5 L taking into account any additional physical activity. The main drink consumed by children should be water, but other unsweetened beverages are also allowed. It is also worth paying attention to vegetables and fruits characterized by high water content, i.e., cucumbers, tomatoes, melons, oranges, or apples. Children must avoid sweetened fruit or carbonated drinks in their daily diet. The child’s daily schedule should include a daily walk or outdoor play. Parents should be encouraged to exercise with their children, adjusting the activity to their abilities and available conditions. Parents should also pay attention to the amount of time children spend in front of a TV or computer screen and limit it to the minimum necessary. A child’s daily diet should be rich in vegetables, fruits, pulses, low- or unprocessed grain products, potatoes, nuts, and foods of animal origin, i.e., meat, dairy products, fish, or eggs, in amounts adapted to the child’s nutritional needs. It is worth introducing children to the habit of eating raw vegetables and fresh fruits as snacks. Children should consume unsaturated fats, that is, those contained in fish, avocados, nuts, and olive oil. Their daily diet should not include saturated fats, i.e., lard, coconut oil, or fatty meat. The diet should include dairy products in the recommended amounts, which provide vitamins D, A, and B2, as well as beneficial fatty acids. The child should eat a variety of lean meats and fish instead of prepared meats that contain a lot of salt and fat. Highly processed foods are contraindicated in a child’s diet, i.e., fast food, various ready-to-eat snacks, frozen pizza and cakes, and pastries, due to their content of trans fatty acids. Daily consumption of table salt should not exceed 5 g, and salt included in the diet should be iodized. Parents should be cautioned to avoid adding salt or sodium-containing condiments when preparing meals. All snacks with high salt or sugar content, i.e., salty sticks or candy, are contraindicated in the child’s diet. To improve the quality of the daily diet, families should be encouraged to prepare home-cooked meals, which tend to be healthier and more nutritious for growing children, compared to restaurant or industrially prepared meals. Additionally, during the COVID-19 pandemic, eating out increases the risk of contracting coronavirus. Children should be involved in choosing food and preparing home-cooked meals together. If there are any problems related to a child’s eating, consult specialists, i.e., a nutritionist, dietician, or psychologist, and seek psychosocial support from family and friends.

Changes in health behavior during the COVID-19 pandemic are important and should be controlled through extensive health promotion and health education activities, as also highlighted in other studies by the authors of this paper [37,38,39]. The author’s study indicates the importance of implementing nutrition education for children and their parents in the context of mental, social, and physical health to reduce the development of chronic non-communicable diseases (such as overweight and obesity).

### Strengths and Limitations

The strengths of the survey are its reliance on a validated questionnaire and its adaptation to Polish conditions, and the performed revalidation. The disadvantage, however, is the small amount of socioeconomic data due to the length of the questionnaire. In addition, dietary habits were assessed, asking about the consumption of sweets, salty snacks, and fast food, as well as the number of meals consumed per day, physical activity, time spent in front of screens, feelings of anxiety and stress, and the factors that respondents felt most influenced them. A strength of our study is the timing of the survey. These were the first 3 months of the most restrictive lockdown, which was introduced in Poland at the very beginning of the COVID-19 pandemic. It was during this time that the lives of Polish children and their parents changed most dramatically. An important limitation of the study is that the data were self-reported, from which anthropometric indices were calculated, which may have resulted in a bias. The questionnaire asked about current weight and height at the time of the survey, from which BMI was calculated and related to centile grids. These variables were not asked in the pre-pandemic aspect, as it was assumed that such data could be skewed, and in the aspect of the present study, it would have been more reasonable to ask about the parent’s subjective assessment of the child’s change or lack of weight change, which is largely influenced by the child’s intensive growth at this developmental age. Another limitation of the study was the lack of socio-demographic data on the parents of the study group of children. In addition, the survey did not take into account several confounding variables that may have influenced respondents’ behavior. It is planned to address this topic at a later date to assess the long-term effects and increase the pool of variables covered.

In the case of the survey conducted, a limitation was the problem of collecting a larger sample group. This may have been due to the large number of surveys sent electronically recently and the fatigue of parent-respondents as to their completion. Nevertheless, the study joins the ranks of works that have marked the role of the pandemic in terms of the daily lives of children, as well as entire families. It also underscores the significant role that dietitians can play in the near term in disseminating up-to-date nutrition knowledge based on EBN (Evidence-Based Nutrition) and thereby facilitating the delivery of professional dietary care and the implementation of effective nutrition interventions for children, which seems particularly important in the post-pandemic period.

## 5. Conclusions

Pandemic COVID-19 affected the eating behavior and lifestyle of children attending primary schools:pandemic COVID-19 changed the body weight of almost half of the children surveyed, with more weight gains than weight losses;there was an increase in the consumption of salty snacks and high-sugar products and sweetened beverages;a consequence of the COVID-19 pandemic was a deterioration in dietary habits through a reduction in children’s intake of essential dietary components, i.e., fruit and vegetables and milk and dairy products.

Subsequently, the pandemic had a significant impact on reduced levels of physical activity and increased time spent in front of a screen. In summary, among the reasons for changes in children’s dietary and lifestyle habits, factors related to the pandemic itself, i.e., social restrictions, restrictions, closure of schools and other facilities, and fear of coronavirus infection, had the greatest impact.

## Figures and Tables

**Figure 1 nutrients-15-01928-f001:**
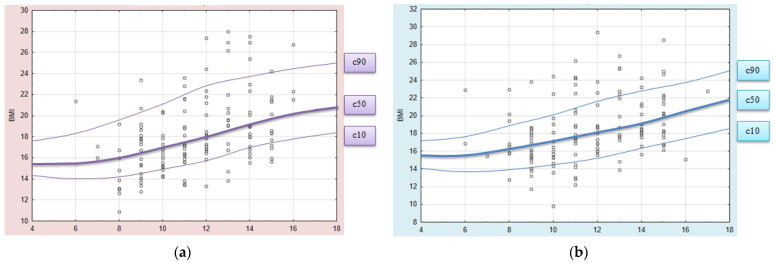
Graph of BMI distribution by age and gender (**a**) Graph for girls; (**b**) Graph for boys.

**Figure 2 nutrients-15-01928-f002:**
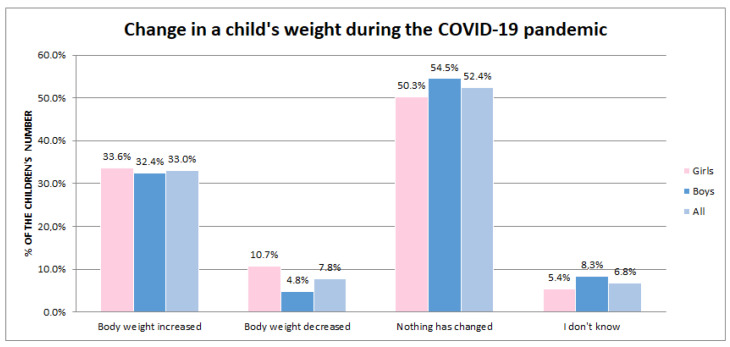
Change in the body weight of the studied children during the COVID-19 pandemic.

**Figure 3 nutrients-15-01928-f003:**
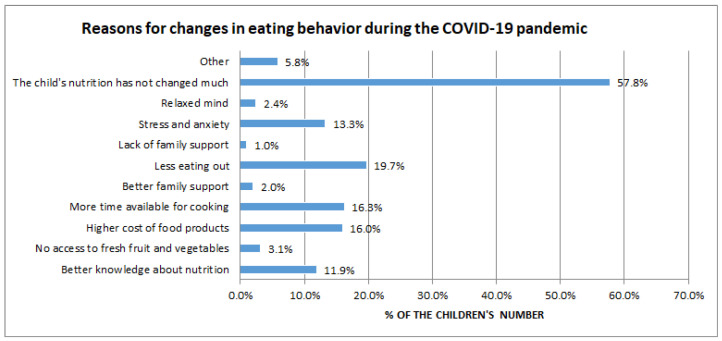
Reasons for changes in eating behavior during the COVID-19 pandemic.

**Figure 4 nutrients-15-01928-f004:**
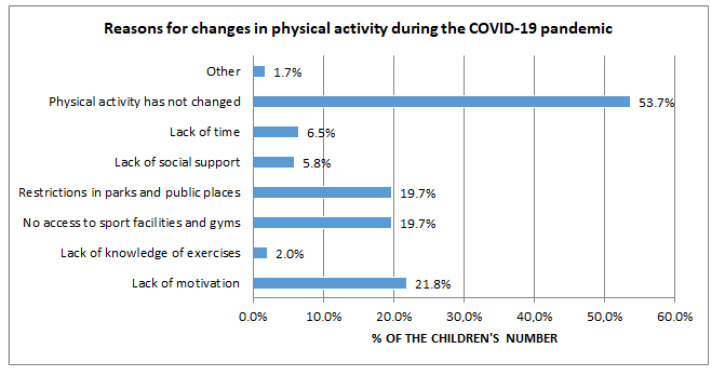
Reasons for changes in physical activity during the COVID-19 pandemic.

**Table 1 nutrients-15-01928-t001:** Characteristics of the study group (N = 294).

Variable		N (%)	*p*-Value
Gender	Girls	149 (50.7)	>0.05 *
	Boys	145 (49.3)	
Grade	1–3 (6–8 y.)	108 (36.7)	
	4–6 (9–11 y.)	99 (33.7)	
	7–8 (12–14 y.)	87 (29.6)	
Place of residence	City	155 (52.7)	
	Country	139 (47.3)	

* There were no statistically significant differences between the groups, assuming that in terms of division, the groups are homogeneous.

**Table 2 nutrients-15-01928-t002:** Level of frequency of consumption of five regular meals per day before and during the COVID-19 pandemic.

Level of Frequency of Consumption of 5 Regular Meals before the COVID-19 Pandemic	Level of Frequency of Consumption of 5 Regular Meals during the COVID-19 Pandemic						
	Never	1–2 Times a Week	3–4 Times a Week	5–6 Times a Week	Daily	Total (%)	*p*-Value (Chi^2^ NW)
Never	5 (62.50%)	2 (25.00%)	0 (0.00%)	1 (12.50%)	0 (0.00%)	8 (100%)	*p* < 0.001
1–2 times a week	2 (11.11%)	10 (55.56%)	3 (16.67%)	1 (5.56%)	2 (11.11%)	18 (100%)	
3–4 times a week	1 (1.69%)	8 (13.56%)	31 (52.54%)	14 (23.73%)	5 (8.47%)	59 (100%)	
5–6 times a week	2 (3.92%)	5 (9.80%)	10 (19.61%)	26 (50.98%)	8 (15.69%)	51 (100%)	
Daily	1 (0.63%)	4 (2.53%)	11 (6.96%)	16 (10.13%)	126 (79.75%)	158 (100%)	

**Table 3 nutrients-15-01928-t003:** The frequency level of fruit and vegetable consumption before and during the COVID-19 pandemic.

The Frequency Level of Fruit and Vegetable Consumption before the COVID-19 Pandemic	The Frequency Level of Fruit and Vegetable Consumption during the COVID-19 Pandemic						
	Never	1–2 Times a Week	3–4 Times a Week	5–6 Times a Week	Daily	Total (%)	*p*-Value (Chi^2^ NW)
Never	5 (83.33%)	1 (16.67%)	0 (0.00%)	0 (0.00%)	0 (0.00%)	6 (100%)	*p* < 0.001
1–2 times a week	1 (2.56%)	26 (66.67%)	9 (23.08%)	3 (7.69%)	0 (0.00%)	39 (100%)	
3–4 times a week	1 (2.08%)	10 (12.99%)	50 (64.94%)	16 (20.78%)	1 (1.30%)	77 (100%)	
5–6 times a week	0 (0.00%)	7 (12.50%)	19 (33.93%)	26 (46.43%)	4 (7.14%)	56 (100%)	
Daily	0 (0.00%)	3 (2.59%)	12 (10.34%)	9 (7.76%)	92 (79.31%)	116 (100%)	

**Table 4 nutrients-15-01928-t004:** Prevalence level of salty snack consumption before and during the COVID-19 pandemic.

Level of Frequency of Salty Snack Consumption before the COVID-19 Pandemic	Level of Frequency of Salty Snack Consumption during the COVID-19 Pandemic						
	Never	1–2 Times a Week	3–4 Times a Week	5–6 Times a Week	Daily	Total (%)	*p*-Value (Chi^2^ NW)
Never	28 (73.68%)	8 (21.05%)	1 (2.63%)	1 (2.63%)	0 (0.00%)	38 (100%)	*p* < 0.001
1–2 times a week	7 (3.63%)	141(73.06%)	35 (18.13%)	9 (4.66%)	1 (0.52%)	193 (100%)	
3–4 times a week	1 (2.08%)	11 (22.92%)	28 (58.33%)	2 (4.17%)	6 (12.50%)	48 (100%)	
5–6 times a week	0 (0.00%)	1 (16.67%)	3 (50.00%)	2 (33.33%)	0 (0.00%)	6 (100%)	
Daily	0 (0.00%)	2 (22.22%)	0 (0.00%)	0 (0.00%)	7 (77.78%)	9 (100%)	

**Table 5 nutrients-15-01928-t005:** Level of frequency of moderate-intensity exercise/sports before and during the COVID-19 pandemic.

Level of Frequency of Exercise/Moderate Intensity Sports before the COVID-19 Pandemic	Level of Frequency of Moderate-Intensity Exercise/Sports during the COVID-19 Pandemic						
	Never	1–2 Times a Week	3–4 Times a Week	5–6 Times a Week	Daily	Total (%)	*p*-Value (Chi^2^ NW)
Never	18 (69.23%)	7 (26.92%)	1 (3.85%)	0 (0.00%)	0 (0.00%)	26 (100%)	*p* < 0.001
1–2 times a week	13 (15.85%)	60 (73.17%)	6 (7.32%)	1 (1.22%)	2 (2.44%)	82 (100%)	
3–4 times a week	3 (3.06%)	38 (38.78%)	44(44.90%)	9 (9.18%)	4 (4.08%)	98 (100%)	
5–6 times a week	1 (2.63%)	5 (13.16%)	17 (44.74%)	13 (34.21%)	2 (5.26%)	38 (100%)	
Daily	3 (6.00%)	9 (18.00%)	10 (20.00%)	5 (10.00%)	23 (46.00%)	50 (100%)	

**Table 6 nutrients-15-01928-t006:** Length of time spent in front of a TV/phone/computer screen before and during the COVID-19 pandemic.

Time Spent in front of TV/Phone/Computer Screen before the COVID-19 Pandemic	Time Spent in front of TV/Phone/Computer Screen during the COVID-19 Pandemic					
	Less Than an Hour	1–2 h	3–4 h	More Than 4 h	Total (%)	*p*-Value (Chi^2^ NW)
Less than an hour	28 (45.90%)	18 (29.51%)	7 (11.48%)	8 (13.11%)	61 (100%)	*p* < 0.001
1–2 h	3 (2.19%)	57 (41.61%)	53 (38.69%)	24 (17.52%)	137 (100%)	
3–4 h	0 (0.00%)	4 (4.94%)	45 (55.56%)	32 (39.51%)	81 (100%)	
More than 4 h	0 (0.00%)	0 (0.00%)	4 (26.67%)	11 (73.33%)	15 (100%)	

**Table 7 nutrients-15-01928-t007:** Length of time spent sleeping before and during the COVID-19 pandemic.

Time Allotted for Sleep before the COVID-19 Pandemic	Time Spent Sleeping during the COVID-19 Pandemic				
	Less Than 6 h	6–8 h	More Than 8 h	Total (%)	*p*-Value (Chi^2^ NW)
Less than 6 h	3 (50.00%)	2 (33.33%)	1 (16.67%)	6 (100%)	*p* < 0.001
6–8 h	12 (8.51%)	120 (85.11%)	9 (6.38%)	141 (100%)	
More than 8 h	0 (0.00%)	35 (23.81%)	112 (76.19%)	147 (100%)	

**Table 8 nutrients-15-01928-t008:** Level of perceived stress/anxiety before and during the pandemic COVID-19.

Level of Perceived Stress/Anxiety of the Pandemic COVID-19	Level of Perceived Stress/Anxiety of the Pandemic COVID-19					
	Did Not Feel It at All	Sometimes	Often	Very Often	Total (%)	*p*-Value (Chi^2^ NW)
Did not feel It at all	44 (55.00%)	29 (36.25%)	6 (7.50%)	1 (1.25%)	80 (100%)	*p* < 0.001
Sometimes	6 (3.00%)	154 (77.00%)	33 (16.50%)	7 (3.50%)	200 (100%)	
Often	0 (0.00%)	3 (27.27%)	7 (63.64%)	1 (9.09%)	11 (100%)	
Very often	1 (33.33%)	1 (3.33%)	0 (0.00%)	1 (33.33%)	3 (100%)	

## Data Availability

Not applicable.

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
