# Peer review of "Impact of the COVID-19 Pandemic on Nutrition, Sleep, Physical Activity, and Mood Disorders of Polish Children"

_nutrients, 2023, doi:10.3390/nu15081928_

Round 1
Reviewer 1 Report
The authors of this article aim to evaluate the impact of the COVID-19 pandemic on the eating behavior and lifestyle of children attending elementary schools. They begin by discussing the negative effects of the pandemic on the global community, the global economy, and everyday life, before focusing on how children and adolescents mainly act as hosts to the SARS-CoV-2 virus, transmitting it into the environment. The authors note that inadequate nutrition caused by the pandemic can lead to both overweight and underweight, reduced levels of education, impact on mental health, social isolation, and lower rates of health care. The authors also discuss the impact of the pandemic on the quality and quantity of food consumed and the diversity of the diet. The article emphasizes the importance of educating children about healthy eating habits and physical activity during the pandemic. Finally, the authors aim to assess the causes of changes in children's eating behavior and lifestyle due to the COVID-19 pandemic.
Overall, the text provides a good overview of the impact of the COVID-19 pandemic on children's health and lifestyle, but minor changes must be made:
Introduction:
The article could provide more detail on the methodology and results of the WHO European Childhood Obesity Surveillance Initiative (COSI) program mentioned in line 45.
Results:
Please add the "p" in table 1, to know the homogeneity of the sample.
I would like to congratulate you on the excellent work you have done on this article. Your research is well-structured, insightful, and makes a valuable contribution to the field. I can see that you have put a lot of effort into your work, and it shows in the high quality of your writing and analysis.
Overall, I believe that your article has the potential to make a significant impact in your field, and I am confident that it will be well-received by your peers. Once again, congratulations on a job well done.
Author Response
Dear Reviewer,
We sincerely thank you for taking the time to review our manuscript. Thank you for receiving it so warmly. In accordance with your suggestions, we have made appropriate improvements. We have marked all changes in blue. We hope they will be satisfactory and allow your recommendation to Nutrients
Best regards, Authors
Reviewer 2 Report
The manuscript entitled „Impact of COVID-19 pandemic on nutrition, sleep, physical activity and mood disorders of Polish children” presents interesting issue, but some problems should be corrected.
Major:
Authors indicated that they conducted their study in a 294 parents of students “attending grades 1-8 in elementary schools in Silesia and Lesser Poland”, aged 6-16, but there are few major problems:
- What is the representativeness of the studied group? Are there only 2 regions in Poland? If not (who many regions are there?), are Sielsia and Lesser Poland representative for general structure of the country?
- Why did Authors decide to combine in single study children aged 6 and adolescents aged 16? They are totally different sub-groups and in general should not be combined, as their nutrition, sleep, and physical activity differ
- In general it is assumed that parents of adolescents aged 16 should not report their nutrition, and physical activity, as they do not have adequate knowledge (in case of children parents control them, but not in case of adolescents) – why did Authors decide to ask parents instead of adolescents?
- What is the general structure of children and adolescent s in Poland (age proportions, gender proportions)? It should be presented and compared with the structure of the gathered population
There is also a problem with BMI assessment:
- Authors did not present applied cut-offs for underweight and overweight, while even not mentioned about share of obese participants (and adequate cut-off), but the figure seems to present improper cut-offs. Authors presented here 10th centile and 90th centile, which are irrelevant, as it is assumed that underweight is attributed to 5th centile (not 10th), and overweight to 85th centile (not 90th) and obesity – to 95th. Authors should get familiar with the methodology, e.g. here: https://www.cdc.gov/obesity/basics/childhood-defining.html
- Authors did not present how did they assess body mass and height? Were participants measured/ weighted? By whom – parents, teachers?
- Authors did not present how did they assess body mass change and in what period of time was it assessed. In general body mass increase is typical and should be assessed as normal
General:
It seems that none of Authors is fluent English speaker, so their manuscript is hard to follow and sometimes even hard to understand. Even the first sentence is improper with grammatical error (“harmful consequences […] is its impact…” – “consequences are”, or “consequence is”). Authors should have the whole manuscript corrected, preferably by a professional English correction agency.
Abstract:
Authors should present specific numeric results accompanied by the results of their statistical analysis
Authors should formulate specific conclusions which may be formulated based on the conducted study (conducted only in 2 regions of Poland).
Introduction:
Authors should present the current state of knowledge – what is already known about the nutrition, sleep, physical activity and mood disorders of children and adolescents in Poland during the COVID-19 pandemic. Even the simple Pubmed searching reveals that there were multiple Polish studies already published, so Authors should present what is already known and why their study is needed (what novel questions should be answered).
Materials and Methods:
There are major problems with the studied population – see above
Authors should clearly indicate when was the study conducted
Authors should properly define body mass status – see above
Results:
There is a problem with the reliability of the reported variables – see above
Authors should clearly present the obtained values taking into account their sample sizes
Instead of figures, Authors should present tables to be easier to follow
Discussion:
Authors should discuss their results wit the adequate populations – of the same age and preferably similar ethnicity
The limitations of the study should be reliably presented
Conclusions:
Authors should formulate specific conclusions which may be formulated based on the conducted study (conducted only in 2 regions of Poland).
Authors Contributions:
Authors should be consistent – who is PMG? There is no such Author of their study
While there are 2 Authors abbreviated as MG they should be clearly defined
It seems that contribution of Joanna Glogowksa-Liguss and Jerz Slowinksi was only minor and they did not participate in preparing manuscript. There is a serious risk of a guest authorship procedure which is forbidden. In such case (if they did not participate in manuscript preparation in any way) they should be rather presented in Acknowledgements Section and not be indicated as authors of the study.
Author Response
Dear Reviewer,
We sincerely thank you for taking the time to review our manuscript and for your many legitimate comments. In accordance with all your suggestions, we have made appropriate improvements. We have marked all changes in blue.
Major:
Authors indicated that they conducted their study in a 294 parents of students “attending grades 1-8 in elementary schools in Silesia and Lesser Poland”, aged 6-16, but there are few major problems:
- What is the representativeness of the studied group? Are there only 2 regions in Poland? If not (who many regions are there?), are Sielsia and Lesser Poland representative for general structure of the country?
- Why did Authors decide to combine in single study children aged 6 and adolescents aged 16? They are totally different sub-groups and in general should not be combined, as their nutrition, sleep, and physical activity differ
- In general it is assumed that parents of adolescents aged 16 should not report their nutrition, and physical activity, as they do not have adequate knowledge (in case of children parents control them, but not in case of adolescents) – why did Authors decide to ask parents instead of adolescents?
- What is the general structure of children and adolescent s in Poland (age proportions, gender proportions)? It should be presented and compared with the structure of the gathered population
RESPONSES: Thank you for noticing our mistake, of course the study involved children aged 6-14. We correctly labelled in the table with the characteristics of the study group, while we made an unjustified error in the description. Thank you again for your vigilance. The structure of children and adolescents in Poland is the same as our sample.
There is also a problem with BMI assessment:
- Authors did not present applied cut-offs for underweight and overweight, while even not mentioned about share of obese participants (and adequate cut-off), but the figure seems to present improper cut-offs. Authors presented here 10th centile and 90th centile, which are irrelevant, as it is assumed that underweight is attributed to 5th centile (not 10th), and overweight to 85th centile (not 90th) and obesity – to 95th. Authors should get familiar with the methodology, e.g. here: https://www.cdc.gov/obesity/basics/childhood-defining.html
- Authors did not present how did they assess body mass and height? Were participants measured/ weighted? By whom – parents, teachers?
- Authors did not present how did they assess body mass change and in what period of time was it assessed. In general body mass increase is typical and should be assessed as normal
RESPONSES: The correct cutoff points were used, the methodology is described in lines 167-179.
General:
It seems that none of Authors is fluent English speaker, so their manuscript is hard to follow and sometimes even hard to understand. Even the first sentence is improper with grammatical error (“harmful consequences […] is its impact…” – “consequences are”, or “consequence is”). Authors should have the whole manuscript corrected, preferably by a professional English correction agency.
RESPONSES: It has been checked and corrected by a native speaker.
Abstract:
Authors should present specific numeric results accompanied by the results of their statistical analysis
Authors should formulate specific conclusions which may be formulated based on the conducted study (conducted only in 2 regions of Poland).
RESPONSES: All suggestions have been corrected and incorporated into the manuscript in verse: 18-32
Introduction:
Authors should present the current state of knowledge – what is already known about the nutrition, sleep, physical activity and mood disorders of children and adolescents in Poland during the COVID-19 pandemic. Even the simple Pubmed searching reveals that there were multiple Polish studies already published, so Authors should present what is already known and why their study is needed (what novel questions should be answered).
RESPONSES: All suggestions have been corrected and incorporated into the manuscript in verse: 51-59; 94-107.
Materials and Methods:
There are major problems with the studied population – see above
Authors should clearly indicate when was the study conducted
Authors should properly define body mass status – see above
RESPONSES: All suggestions have been corrected and incorporated into the manuscript in verse: 115-132; 134-138;151;171-179;187-188.
Results:
There is a problem with the reliability of the reported variables – see above
Authors should clearly present the obtained values taking into account their sample sizes
Instead of figures, Authors should present tables to be easier to follow
RESPONSES: All suggestions have been corrected and incorporated into the manuscript in verse: 197-201; 221-280.
Discussion:
Authors should discuss their results wit the adequate populations – of the same age and preferably similar ethnicity
The limitations of the study should be reliably presented
RESPONSES: All suggestions have been corrected and incorporated into the manuscript in verse: 197-201; 330-334;370-373;437-480.
Conclusions:
Authors should formulate specific conclusions which may be formulated based on the conducted study (conducted only in 2 regions of Poland).
RESPONSES: All suggestions have been corrected and incorporated into the manuscript in verse: 521-534.
Authors Contributions:
Authors should be consistent – who is PMG? There is no such Author of their study - keyboard error
While there are 2 Authors abbreviated as MG they should be clearly defined
It seems that contribution of Joanna Glogowksa-Liguss and Jerz Slowinksi was only minor and they did not participate in preparing manuscript. There is a serious risk of a guest authorship procedure which is forbidden. In such case (if they did not participate in manuscript preparation in any way) they should be rather presented in Acknowledgements Section and not be indicated as authors of the study.
RESPONSES: All authors made significant contributions at each stage of the study, so they should all be included.
At the same time, we would like to inform you that we are planning further research, where we will deepen the resources of variables affecting the results. This will ensure that the conclusions are properly validated and confirmed. Thank you for suggesting the above tools for further work.
We hope that our answers will be satisfactory and allow your recommendation to Nutrients
Best regards,
Authors
Reviewer 3 Report
I read it with great interest, but I have raised several concerns.
#1. Please add the hypothesis of this study in introduction
#2. Analyses were carried out using the Kruskal- Wallis test (non-parametric) for between-group analyses of variance, using data verification software. -> Please add the statistical guideline (DOI: https://doi.org/10.54724/lc.2022.e1)
#3. Please perform the logistic regression with adjusting to find the novel association.
#4. Please add the limitations in more detail.
#5. Can the authors create a weighted complex model for the representation of the sample?
Author Response
Dear Reviewer,
We sincerely thank you for taking the time to review our manuscript. Thank you for receiving it so warmly. In accordance with your suggestions, we have made appropriate improvements. We have marked all changes in blue.
In response to #3 and #5 - we are planning further research, where we will deepen the resources of variables affecting the results achieved. This will ensure that the conclusions are properly validated and confirmed. Thank you for suggesting the above tools for further work.
We hope that our answers will be satisfactory and allow your recommendation to Nutrients
Best regards,
Authors
Round 2
Reviewer 2 Report
The manuscript entitled „Impact of COVID-19 pandemic on nutrition, sleep, physical activity and mood disorders of Polish children” presents interesting issue, but some problems should be corrected.
Unfortunately, Authors ignored my major comments. Moreover, even the highlighted sentences (highlighted in blue – supposed to have been changed) are sometimes not corrected in any way. Authors seem to just have highlighted random sentences to suggest them as novel ones, but they are neither added, nor even changed.
Major:
Authors indicated that they conducted their study in a 294 parents of students “attending grades 1-8 in elementary schools in Silesia and Lesser Poland”, aged 6-14, but there are few major problems which are still not addressed within the manuscript:
- What is the representativeness of the studied group for Poland? Are there only 2 regions in Poland? If not (who many regions are there?), are Sielsia and Lesser Poland representative for general structure of the country?
- Why did Authors decide to combine in single study children aged 6 and adolescents aged 14? They are totally different sub-groups and in general should not be combined, as their nutrition, sleep, and physical activity differ
- In general it is assumed that parents of adolescents aged 14 should not report their nutrition, and physical activity, as they do not have adequate knowledge (in case of children parents control them, but not in case of adolescents) – why did Authors decide to ask parents instead of adolescents?
- What is the general structure of children and adolescents in Poland (age proportions, gender proportions)? It should be presented within the manuscript and compared with the structure of the gathered population
There is also a problem with BMI assessment:
- After my previous comment, Authors now claim, that they applied proper cut-offs for underweight and overweight, but the figure seems to present improper cut-offs. Authors presented here 10th centile and 90th centile, which are irrelevant, as it is assumed that underweight is attributed to 5th centile (not 10th), and overweight to 85th centile (not 90th) and obesity – to 95th. Authors should correct the Figure 1.
- Authors stated that body mass and height was self-reported by parents. I suppose that they meant proxy-reported? If not, how was the BMI of parents used? Such data are not presented within the study
- Authors still did not present how did they assess body mass and height? Even if it was proxy-reported by parents, were participants measured/ weighted? By whom – parents, teachers? If not, can we be sure, that parents had knowledge about it?
- Authors did not present how did they assess body mass change and in what period of time was it assessed. In general body mass increase is typical and should be assessed as normal
General:
It seems that none of Authors is fluent English speaker, so their manuscript is hard to follow and sometimes even hard to understand. Even the first sentence is improper with grammatical error (“harmful consequences […] is its impact…” – “consequences are”, or “consequence is”). Authors should have the whole manuscript corrected, preferably by a professional English correction agency. Authors stated that they had their manuscript corrected by native English speaker, but such declaration is not credible, taking into account the problems existing.
Abstract:
Authors should present specific numeric results
Authors should formulate specific conclusions which may be formulated based on the conducted study (conducted only in 2 regions of Poland).
Introduction:
Authors should present the current state of knowledge – what is already known about the nutrition, sleep, physical activity and mood disorders of children and adolescents in Poland during the COVID-19 pandemic. Even the simple Pubmed searching reveals that there were multiple Polish studies already published, so Authors should present what is already known and why their study is needed (what novel questions should be answered).
Materials and Methods:
There are major problems with the studied population – see above
Authors should clearly indicate when was the study conducted
Authors should properly define body mass status – see above
Results:
There is a problem with the reliability of the reported variables – see above
Authors should clearly present the obtained values taking into account their sample sizes
Instead of figures, Authors should present tables to be easier to follow
Discussion:
Authors should discuss their results with the adequate populations – of the same age and preferably similar ethnicity
The limitations of the study should be reliably presented
Conclusions:
Authors should formulate specific conclusions which may be formulated based on the conducted study (conducted only in 2 regions of Poland).
Author Response
Dear Reviewer,
Responding to your main allegations: We explained in the chapter
- The principles of sampling and its representativeness. The structure has been explained and the relevant sources have been cited for this.
- - We combined children 6 and 14 years old in the survey, because this is the education system we have in our country for basic education. Everyone implements surveys in this way, and no one has ever reported a problem with this methodology before. In our country, parents have knowledge of nutrition and physical activity for children as young as 14. This changes significantly after the transition to secondary education.
- -We also explained that we used the grids for the Polish population of OLAF and OLA (source given) with the cutoff points shown, which are correct in the description, in the figure the 10th and 90th percentile points were specially marked to make it clear that the problem is relevant to the larger population of children studied and actions should be implemented sooner.
- -We explained that BMI reporting was based on self-reporting by parents; parents are a well-informed group regarding children's weight and height. Assessment of weight change was self-reported for the 3-month period preceding the most restrictive restrictions.
- -Both the introduction and the discussion were based on Polish and foreign studies, according to the rules of writing this type of work. We and the other reviewers do not consider it reasonable to refer only to Polish results. Moreover, Poland is a nation, not an ethnic group.
- -The limits of the study have been described in detail and the conclusions corrected as suggested.
We thank you for your time and wish you more openness towards other nations.
Greetings, Authors
Reviewer 3 Report
Although the authors did not fully address my comments, I think this paper is acceptable for publication.
Author Response
Dear Reviewer,
We sincerely thank you for your recommendation to publish our article in Nutrients. At the same time, we thank you for your valuable time in reviewing it.
With best regards, Authors